# Coupling of Finite Element Method and Peridynamics to Simulate Ship-Ice Interaction

**Renwei Liu [1], Yanzhuo Xue [2,***] and Xikui Lu [2]**

1   School of Naval Architecture and Ocean Engineering, Jiangsu University of Science and Technology, Zhenjiang 212100, China
2   College of Shipbuilding Engineering, Harbin Engineering University, Harbin 150001, China
*   Correspondence: xueyanzhuo@hrbeu.edu.cn

**Abstract:** In this work, the finite element method (PD-FEM) coupling strategy is used to simulate ship-ice interaction. Two numerical benchmark tests are selected to validate the coupling approach and its program. During the ice-breaking process simulation, the generation and propagation of radial and circular cracks in level ice are modeled and phenomena such as the shedding of wedge ice, flipping of brash ice, and cleaning of the channel are observed to be broadly consistent with experimental observation. The influence of ship speed and ice thickness on the ice load are investigated and analyzed. The ice load obtained from the numerical simulations is in general agreement with that given by Lindqvist's empirical formula. The boundary effect on the crack path can also be avoid with the current coupling method.

**Keywords:** level ice; ship-ice interaction; PD-FEM coupling approach; ice-breaking process; ice load

## 1. Introduction

In recent years, global climate change and the melting of ice in Arctic regions has raised the possibility of exploiting Arctic resources and opening an Arctic channel [1,2]. The exploitation of resources and scientific research in Arctic regions rely on icebreakers to open the necessary routes [3–5]. Therefore, it is great significant to simulate the icebreaking scenarios and calculate the ice load of ship–ice interaction, and it helps in improving the design and safe navigation of icebreakers. The ship-ice interaction scenarios are studied with full-scale tests, model tests, theoretical analyses, and numerical simulations. For full-scale testing, the results are reliable, but the associated cost is high. Model test is a promising candidate to study the ship–ice interaction [6–9]. However, compared with full-scale tests, models have many uncertainties, and can be expensive and time-consuming [10,11]. Theoretical analysis is still challenging in some cases, such as dealing with complicated structures [12]. Fortunately, numerical methods to study ship-ice interactions do not need to consider the structure complexity, and are not restricted by factors such as geography, cost, and time, and have been shown to be both efficient and accurate, both in theoretical research and engineering application [13–16]. Finite element method (FEM) was successfully applied to estimate the strength of ship structure problems [10,17,18]. The discrete element method (DEM) to calculate ice loads for offshore structures and ships [19–22]. Smoothed-particle hydrodynamics method (SPH) was adopted in the ice field to simulate the ice-structure interaction dynamics [23,24], and other methods [25,26].

In recent years, a mesh-free method of peridynamics was proposed [27]. This reformulation of the classical continuum mechanics is a non-local theory that does not assume the spatial differentiability of displacement fields. Based on integrodifferential equations, peridynamics can deal with discontinuous displacement fields. Therefore, it can simulate spontaneous crack nucleation and propagation, and can be used to simulate the ice-breaking and calculate ice loads [28–37]. However, as a non-local theory computational efficiency of peridynamics is far lower than that of FEM, especially for engineering applications like

ship-ice interaction. To improve its computational efficiency, researchers have coupled peridynamics with FEM. Macek and Silling [38] proposed the PD-FEM coupling approach and implemented peridynamics in a commercial finite element analysis code, Liu et al. [39] introduced interface elements to calculate the coupling force in a PD-FEM approach, and Lee et al. [40] proposed a coupled PD-FEM approach to analyze impact fractures. To date, the advantages of combining PD with FEM have been demonstrated in applications to concrete and composite materials, but PD-FEM coupling has not been used to deal with the ship–ice interaction. In this work, the coupling strategy proposed by Liu et al. is employed for its easy to implement and robust theory foundation.

The following work is organized as, peridynamics theory and PD-FEM coupling scheme is introduced in Sections 2 and 3, respectively. The proposed coupling approach is verified with both dynamic and static cases in Section 4. The ship-ice interaction is simulated in Section 5. Conclusion is drawn in Section 6.

## 2. Peridynamics Framework

Peridynamics assumes that the continuum body is composed of small particles. Each particle interacts with other particles within a finite distance $\delta$ called the horizon. The pairwise interaction between two particles exists despite they are not in contact. This physical interaction is referred to as a bond, which in some way has a close analogy to a mechanical spring. In bond-based peridynamics, the kinetic equation of particle $\mathbf{x}$ in the reference configuration at time $t$ is

$$\rho\ddot{\mathbf{u}}(\mathbf{x},t) = \int_{H_{\mathbf{x}}} \mathbf{f}\big(\mathbf{u}(\mathbf{x}',t) - \mathbf{u}(\mathbf{x},t), \mathbf{x}' - \mathbf{x}\big)dV_{\mathbf{x}'} + \mathbf{b}(\mathbf{x},t)\|\mathbf{x}' - \mathbf{x}\| \leq \delta \tag{1}$$

where $H_{\mathbf{x}}$ is the domain of integration within the horizon of particle $\mathbf{x}$, $\mathbf{u}$ is the displacement vector field, and $\mathbf{b}$ is the body force density. $\rho$ is the mass density, and $\mathbf{f}$ is a pairwise force density function defined as the force per unit volume that particle $\mathbf{x}'$ exerts on particle $\mathbf{x}$, which contains all the constitutive information of the materials.

To simplify the notation, the relative position in the initial configuration and its relative displacement are denoted as $\xi = \mathbf{x}' - \mathbf{x}$ and $\eta = \mathbf{u}(\mathbf{x}',t) - \mathbf{u}(\mathbf{x},t)$, respectively. Therefore, the relative position of the two interacting particles at $t$ in the current configuration is $\xi + \eta$ and the pairwise force density function can be described as $\mathbf{f}(\eta,\xi)$.

For the prototype micro-elastic brittle (PMB) material defined by Silling and Askari [41], the pairwise force density function can be expressed as

$$\mathbf{f}(\eta,\xi) = \frac{\xi + \eta}{|\xi + \eta|}cs\mu(t,\eta,\xi)\forall\eta,\xi \tag{2}$$

where $c = 12E/\pi\delta^4$ is the micro modulus, $E$ is Young's modulus, and $s(\eta,\xi)$ is denoted as the stretch of the bond, which can be defined as

$$s = \frac{|\xi + \eta| - |\xi|}{|\xi|} \tag{3}$$

When the deformation stretch $s$ exceeds a limit $s_0$ (described as the critical stretch for failure), the bond between the two particles breaks and no pairwise force remains. The term $\mu(t,\eta,\xi)$ is a history-dependent scalar-valued function, which is introduced to represent the bond failure of two particles. This can be defined as

$$\mu(t,\eta,\xi) = \begin{cases} 1, & s < s_0 \\ 0, & s \geq s_0 \end{cases} \tag{4}$$

Accordingly, the level of damage is illustrated by the local damage at one particle, defined as

$$\varphi(\mathbf{x}, t) = 1 - \frac{\int_{H_x} \mu(t, \mathbf{\eta}, \mathbf{\xi}) dV_\xi}{\int_{H_x} dV_\xi} \tag{5}$$

When solving the elastic problem in which the damage is not considered, the critical stretch can be set to infinity. Dealing with the damage problem, the value of $s_0$ can be obtained from the energy release rate.

### 3. Coupling of PD-FEM

*3.1. Coupling Scheme*

The PD-FEM coupling approach proposed by Liu et al. [39] is adopted and presented. The coupling scheme is as follows: the domain to be solved is partitioned into FEM subregions, which are modeled as a non-failure area and a PD subregion containing the area expected to be damaged. An interface element is introduced to bridge from the FEM subregion to the PD subregion. The interface element contains several peridynamic nodes for calculating the coupling forces, which are the interaction forces between embedded peridynamics nodes and peridynamics nodes outside the interface element. The coupling scheme is illustrated in Figure 1.

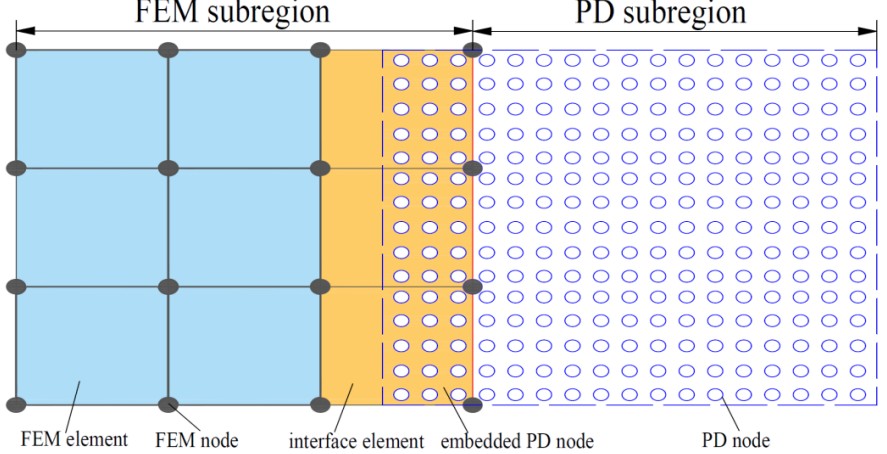

**Figure 1.** PD-FEM coupling scheme.

To implement the coupling scheme, interfaces between the peridynamics subregion and the FEM subregion should be defined prior to analysis. The coupling forces on embedded nodes are divided between the FEM nodes on the interface segment, as shown in Figure 2, by

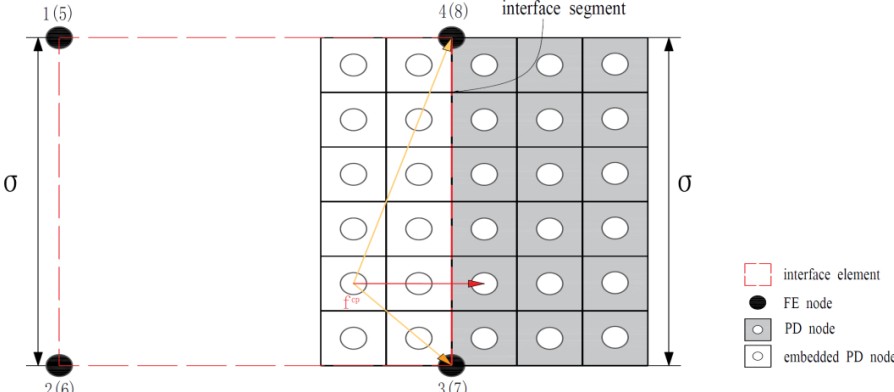

**Figure 2.** Coupling scheme that divides a coupling force $f_p$ to FEM nodes on the interface segment.

$$f_{i,cp} = N_i(\xi', \eta', \psi') f_p \tag{6}$$

where $f_{i,cp}$ is the force of the FEM nodes on the interface segment, $N_i$ is the shape function on the interface segment, $f_p$ is the coupling force on the embedded nodes, $(\xi', \eta', \psi')$ are the natural coordinates of the projection of an embedded node onto the interface segment, $i$ is the number of FEM nodes on the interface segment, and m is the total number of embedded nodes. Note that for FEM nodes that are not on the interface segment, $f_{i,cp} = 0$.

For the FEM subregion, the equation of motion for the FEM nodes is written as

$$M_i \ddot{u}_i^n = F_{i,ext} + F_{i,int} \tag{7}$$

where $F_{i,ext}$ is the external force and the internal force is given by FEM nodes on segment FEM nodes not on segment,

$$F_{i,int} = f_{i,fem} + f_{i,cp} = \begin{cases} \left[\sum_e K^{(e)} u^{(e)}\right]_i + f_{i,cp}, \text{ FEM nodes on segment} \\ \left[\sum_e K^{(e)} u^{(e)}\right]_i, \quad \text{FEM nodes not on segment} \end{cases} \tag{8}$$

The displacements of the embedded peridynamics nodes are determined by

$$u_{ep} = \sum_{i=1}^{8} N_i(\xi, \eta, \psi) u_i \ i = 1, \dots, 8 \tag{9}$$

where $(\xi, \eta, \psi)$ are the natural coordinates of an embedded peridynamics node in the interface element and $u_i$ is the nodal displacement of an interface element.

### 3.2. Numerical Implementation

The peridynamics equation of motion after discretization is written as

$$\rho \ddot{\mathbf{u}}_i^n = \sum_{j=1}^{m} \mathbf{f}\left(\mathbf{u}_j^n - \mathbf{u}_i^n, \mathbf{x}_j - \mathbf{x}_i\right) V_j + \mathbf{b}_i^n \tag{10}$$

For the FEM subregion, the equation of motion for the FEM nodes is written as

$$M_i \ddot{u}_i^n = F_{i,ext}^n + F_{i,int}^n \tag{11}$$

where $n$ denotes the number of time steps. The displacement of node $i$ can be obtained by approximating the acceleration in Equations (12) and (13) using an explicit central difference formula

$$\ddot{\mathbf{u}}_i^n = \frac{\mathbf{u}_i^{n+1} - 2\mathbf{u}_i^n + \mathbf{u}_i^{n-1}}{\Delta t^2} \tag{12}$$

$$\mathbf{u}_i^{n+1} = \begin{cases} \frac{\Delta t^2}{\rho} \left[\sum_{j=1}^{m} \mathbf{f}\left(\mathbf{u}_j^n - \mathbf{u}_i^n, \mathbf{x}_j - \mathbf{x}_i\right) V_j + \mathbf{b}_i^n\right] + 2\mathbf{u}_i^n - \mathbf{u}_i^{n-1}, \text{for PD nodes} \\ \frac{\Delta t^2}{\rho} \left[F_{i,ext}^n - F_{i,int}^n\right] + 2\mathbf{u}_i^n - \mathbf{u}_i^{n-1}, \text{for FEM nodes} \end{cases} \tag{13}$$

where $\Delta t$ is the size of the time step. A stability condition derived by Silling and Askri [41] can be used to determine the time step size, $\Delta t$ as

$$\Delta t < \sqrt{\frac{2\rho}{\sum_{j=1}^{m} \frac{c}{|(x_p - x_i)|} V_j}} \tag{14}$$

Moreover, for the PD particles, the horizon size $\delta$ has a significant influence on the accuracy of the numerical simulations. The horizon size can be selected using the scale characteristics of the simulated object. In practice, $\delta = 3\Delta x$ usually works well [42]. Therefore, the horizon size is set to $\delta = 3\Delta x$.

The numerical code for the proposed PD-FEM coupling approach is compiled using Fortran 90. A flowchart of the PD-FEM coupling approach is shown in Figure 3.

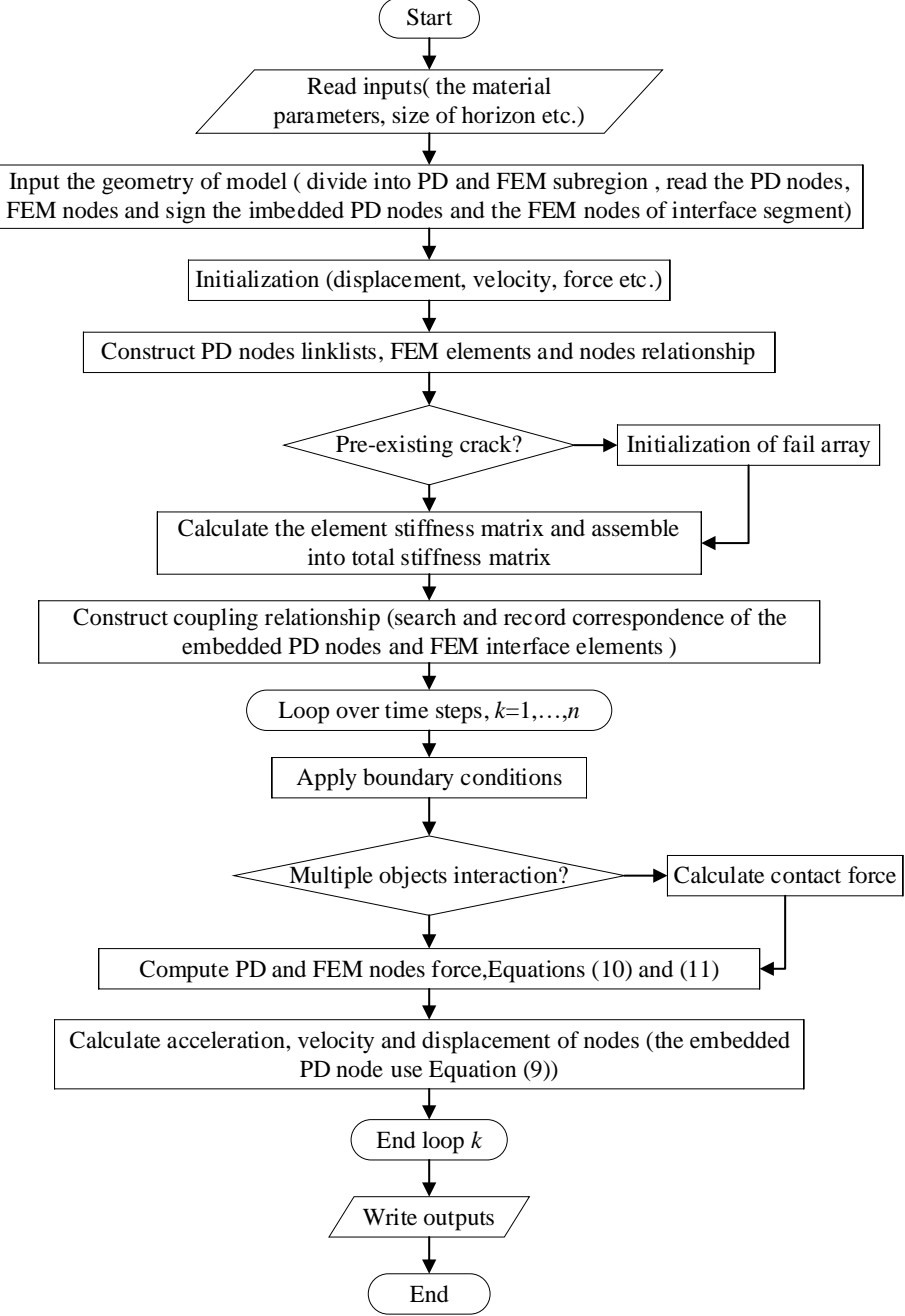

**Figure 3.** Flow chart of the FEM-PD coupling scheme.

## 4. Validation of PD-FEM Coupling Approach

### 4.1. Bending Deformation of Cantilever Beam

A three-dimensional cantilever beam subjected to a transverse loading of $F = 0.64$ N at the free end is examined, and the solutions given by the proposed coupling approach are compared with the FEM solutions. Because the bending deformation of a cantilever

beam is a quasi-static problem, and to achieve a quantitative quasi-static calculation, the dynamic relaxation method is introduced to peridynamics [43].

The cantilever beam is 8 mm long, 2 mm wide, and 2 mm thick with Young's modulus of 1.0 GPa, Poisson's ratio of 0.25, and a density of 900 kg/m$^3$. Figure 4 shows the PD-FEM coupling model of this cantilever beam, which is partitioned into one FEM subregion and one PD subregion. The FEM subregion consists of 16 hexahedral elements of size 1 mm × 1 mm × 1 mm, whereas the PD subregion is discretized uniformly into 16 × 8 × 8 = 1024 particles with the grid spacing $\Delta x = 0.25$ mm and horizon size $\delta = 3\Delta x$. Three layers of peridynamics nodes are embedded in four interface elements for the coupling force calculations.

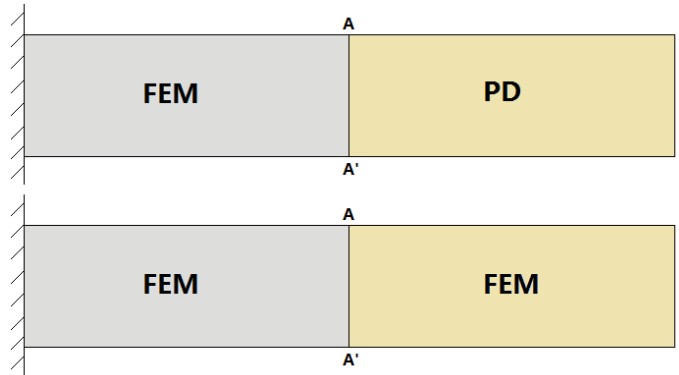

**Figure 4.** PD-FEM coupling model of the cantilever beamThe simulated deflection at the free end of the beam and the coupling force are presented in Table 1. The deflection and force obtained from the proposed coupling approach are very close to the FEM results (error less than 2%), which indicates that the coupling approach transfers the force accurately.

**Table 1.** Simulation results of bending beam.

|  | **PD-FEM** | **FEM** | **Errors** |
|---|---|---|---|
| Deflection | $8.0492 \times 10^{-5}$ m | $8.19 \times 10^{-5}$ m | 1.74% |
| Force | $-0.6327$ N | $-0.64$ N | 1.14% |

Figure 5 shows the change in deflection along the central line of the beam. The displacement curve obtained by the numerical simulation is in good agreement with that of FEM, and its smoothness verifies the displacement coordination of the coupling approach. From Figure 5, it can be found that the ratio of the characteristic scale to the horizon size should be no less than 1 to obtain an acceptable result, however, more cases may be need to achieve this conclusion. The above results prove that the coupling algorithm achieves good accuracy and displacement coordination in the calculation of bending deformation, verifying the correctness of the proposed PD-FEM coupling approach in static problem.

### 4.2. Failure of 2D Plate with Central Crack

Mode-I crack is selected to simulate crack initiation and propagation along the plate. A 50 mm × 50 mm square plate with a 10 mm central pre-crack is stretched from both ends at a velocity of 50 mm/s, as shown in Figure 6.

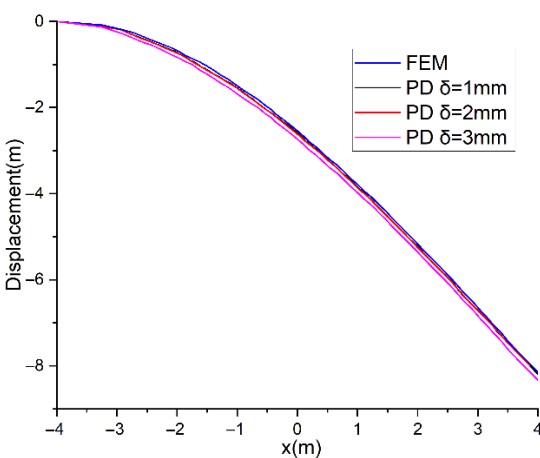

**Figure 5.** Vertical displacement on central axis of the beam.

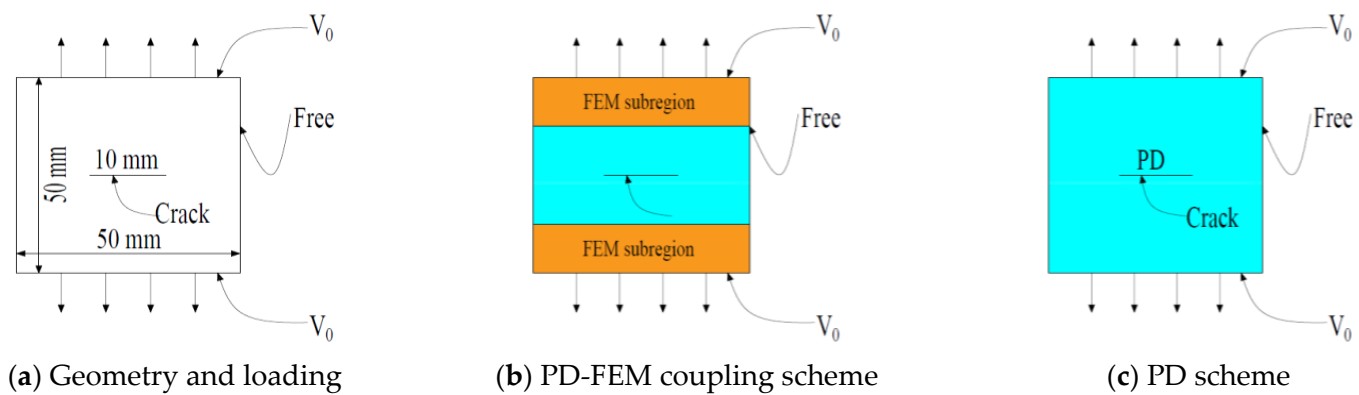

(**a**) Geometry and loading     (**b**) PD-FEM coupling scheme     (**c**) PD scheme

**Figure 6.** Geometry and loading condition of a plate with a central crack.

The PMB material properties used in this example are as follows: Young's modulus is $E$ = 192 GPa, Poisson's ratio is $v$ = 0.33, mass density is 8000 kg/m³, and the critical stretch $s_0$ is 0.04472. These material parameters, geometry, and loading conditions are the same as the simulation example reported by Madenci and Oterkus [42]. For the coupling model, the plate is partitioned into one PD subregion and two FEM subregions (see Figure 6). The PD zone contains 100 × 200 = 20,000 particles, which are discretized regularly with a grid spacing of $\Delta x$ = 0.25 mm and a horizon size of $\delta = 3\Delta x$. The FEM parts are composed of 32 four-node rectangular elements of size 6.25 mm × 6.25 mm. The interface region has three additional layers of peridynamics nodes (total of 2 × 3 × 200 = 1200 nodes). The time step is $\Delta t = 3.34 \times 10^{-8}$ s, which satisfies the stable time step condition.

For comparison, the PD solution is considered, as shown in Figure 6. The model size and load conditions are the same as for the coupling model, and the region is discretized regularly into 200 × 200 = 20,000 nodes with a grid spacing of $\Delta x = 0.25$ mm and horizon size of $\delta = 3\Delta x$. In the velocity boundary area, three virtual boundary layers are added, each with 3 × 200 = 600 nodes.

Figure 7 shows numerical simulation results of crack tracks using the PD-FEM coupling model and PD model (see Figure 7). Crack paths obtained from the proposed coupling method resemble the mode-I failure in brittle material, and are in good agreement with those obtained from the PD method. These results are similar to those reported by Madenci and Oterkus [42]. The displacements along the *y*-axis obtained from the PD-FEM and PD methods are plotted in Figure 8. We can find that they are in close agreement. Therefore, the PD-FEM coupling method can simulate crack propagation well, which verifies the correctness of the coupling approach in dynamic conditions.

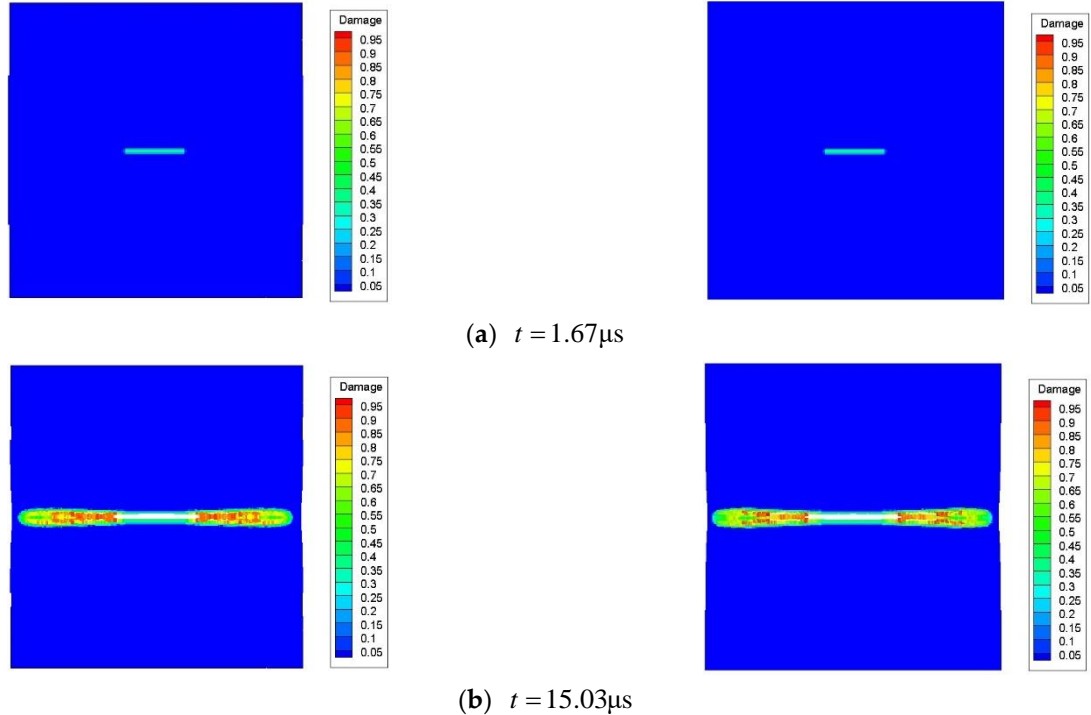

(**a**)  $t = 1.67\mu s$

(**b**)  $t = 15.03\mu s$

**Figure 7.** Crack propagation simulations, (**left**) FEM-PD coupling model and (**right**) PD model.

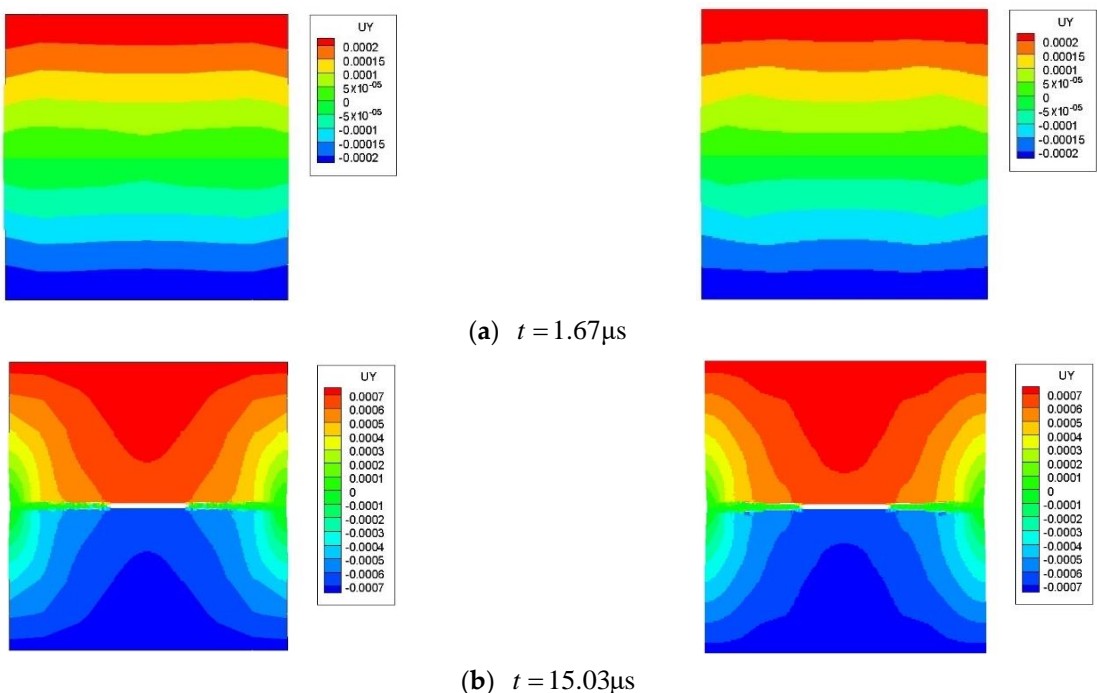

(**a**)  $t = 1.67\mu s$

(**b**)  $t = 15.03\mu s$

**Figure 8.** Vertical displacement simulations, (**left**) FEM-PD coupling model; (**right**) PD model.

## 5. PD-FEM Simulation of Icebreaker Navigation in Ice Level

*5.1. Numerical Simulation*

5.1.1. Ice Constitutive Model and Failure Criterion

Ice is a complex material that is affected by factors such as temperature, porosity, and grain size. Several laboratory tests have analyzed the brittle strength and failure patterns of ice, as well as the characteristics of ice–structure interactions [44,45]. The

compressive strength and tensile strength of ice varies, with the compressive strength being around 3–4 times the tensile strength. The ductile–brittle transition of ice is another challenging topic, and ice shows different behavior at different strain rates (i.e., loading rate), as discussed by Schulson [45].

At low-speed loading rates, ice behaves as a ductile material, whereas at high-speed strain rates, the ice presents the characteristics of a linear elastic material, with a brittle mode when damage occurs. Therefore, ice is regarded as an elastic brittle material (PMB material in bond-based peridynamics) when interacting with an icebreaker, as the relatively high speed of icebreaker vessels corresponds to a high strain rate in the ice. Hence, a reasonable linear elastic constitutive model of ice is established for the bond-based peridynamics. Mechanical ice tests conducted by Schulson can be used to demonstrate the rationality of this linear elastic constitutive model.

In bond-based peridynamics, the bond force represents stress and the bond stretch represents strain. Let $s_t = s_0$ be the critical bond stretch and $|s_c| = 4s_0$ be the critical bond compression. If the bond stretch exceeds $s_t$ in the tensile case or $s_c$ in the compressive case, the bond will break and the bond force becomes zero.

Based on the bond-based peridynamics theory [42], the critical bond stretch in 3D cases is defined as

$$s_0 = \sqrt{\frac{5\pi G_0}{18E\delta}} \tag{15}$$

where $G_0$ is the energy release rate, which reflects the resistance of a material to crack propagation and can be derived from fracture mechanics. Linear elastic fracture mechanics is based on linear elastic theory and is applicable to brittle fractures. From the perspective of energy conservation, the condition for crack propagation is

$$G_0 \geq G_C \tag{16}$$

where $G_C$ is the energy absorption rate. The primary fracture mode is tensile failures, because its compression strength is 3–4 times of tensile strength. $G_0$ can be expressed as

$$G_0 = \frac{K_I^2}{E} \tag{17}$$

where $K_I$ is the fracture toughness, which reflects the resistance of a material to brittle fracture and can be measured experimentally. Therefore, the critical stretch can be calculated as

$$s_0 = \sqrt{\frac{5\pi K_I^2}{18E^2\delta}} \tag{18}$$

5.1.2. The Gravity and Buoyancy Model of Ice

The gravity and buoyancy of the ice are in balance in still water. when interacting with a ship, the ice will deviate from its equilibrium position as the gravity and buoyancy become unbalanced. To simplify the influence of gravity and buoyancy, a body force density $b_z$ is introduced.

If a particle is completely under the waterline, we have

$$b_z = -g\rho_i + g\rho_w \tag{19}$$

If a particle is completely above the waterline, we have

$$b_z = -g\rho_i \tag{20}$$

For other particles, we have

$$b_z = -g\rho_i + g\rho_w l_w / d \tag{21}$$

where $\rho_i$ is the density of ice, $\rho_w$ is the density of water, $l_w$ is the length of particles immersed in water, and $d$ is the size of particles.

5.1.3. Ship-Ice Contact Model

The hull is modeled with FEM. The ice sheet contacting with the hull is modeled with peridynamics. Therefore, the ship–ice contact can be transformed into the interaction between peridynamics and the FEM models. In this case, the contact model developed by Liu [30] is introduced to calculate the contact force between the triangular elements of FEM and the particles of PD. To determine contact between a particle and a triangular element, some specific points must be defined (see Figure 9), namely the vertexes of the triangular element $A$, $B$, $C$ and the particle $P$.

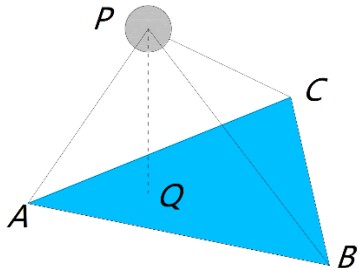

**Figure 9.** Diagram of ship-ice search model.

The determination of contact is divided into two stages. In the first stage, we estimate whether the distance between the particle and the plane of triangle $ABC$ is less than the particle radius $r = \Delta x / 2$. The distance is calculated with Equation (22), and the critical distance is defined by Equation (23)

$$|PQ| = PA \cdot \frac{BA \times CB}{|BA \times CB|} \tag{22}$$

$$|PQ| < r \tag{23}$$

where $Q$ is the projection of $P$ onto the plane of triangle $ABC$.

If the criterion in the first stage is not satisfied, the particle $P$ does not contact the triangle element. When the first stage criterion is satisfied, it is necessary to further decide whether the projection point $Q$ of $P$ is inside the triangular region $ABC$, and this is the second stage. The centroid of the particle is checked to determine whether it is inside the triangle.

For any point $Q$ in the plane $ABC$, the vector $\boldsymbol{d} = AQ$ can be expressed by two non-parallel vectors $\boldsymbol{a} = AC$ and $\boldsymbol{b} = AB$ in plane $ABC$ as

$$\boldsymbol{d} = u\boldsymbol{a} + v\boldsymbol{b} \tag{24}$$

where the coefficients $u$ and $v$ are defined as

$$u = \frac{(\boldsymbol{b} \cdot \boldsymbol{b})(\boldsymbol{d} \cdot \boldsymbol{a}) - (\boldsymbol{b} \cdot \boldsymbol{a})(\boldsymbol{d} \cdot \boldsymbol{b})}{(\boldsymbol{b} \cdot \boldsymbol{b})(\boldsymbol{a} \cdot \boldsymbol{a}) - (\boldsymbol{b} \cdot \boldsymbol{a})(\boldsymbol{a} \cdot \boldsymbol{b})} \tag{25}$$

$$v = \frac{(\boldsymbol{a} \cdot \boldsymbol{a})(\boldsymbol{d} \cdot \boldsymbol{b}) - (\boldsymbol{a} \cdot \boldsymbol{b})(\boldsymbol{d} \cdot \boldsymbol{a})}{(\boldsymbol{a} \cdot \boldsymbol{a})(\boldsymbol{b} \cdot \boldsymbol{b}) - (\boldsymbol{a} \cdot \boldsymbol{b})(\boldsymbol{b} \cdot \boldsymbol{a})} \tag{26}$$

If the projection point $Q$ is inside the triangular element $ABC$, the two coefficients must satisfy conditions, as

$$\begin{aligned} u &\geq 0 \\ v &\geq 0 \\ u + v &\leq 1 \end{aligned} \tag{27}$$

If the criteria in these two stages are satisfied, the particle $P$ is in contact with the triangular element. The contact force is defined by the repelling short-range force. The force between two particles is given as

$$f = -\frac{PQ}{|PQ|}\min\left\{0, c_{sh}\left(\frac{|PQ|}{r} - 1\right)\right\} \tag{28}$$

where $c_{sh}$ is the short-range force coefficient, can be choose as $c_{sh} = 5c$.

### 5.1.4. Numerical Model

This section describes numerical simulations of icebreaker navigation in level ice. Numerical models for the icebreaker and the level ice are illustrated in Figure 10.

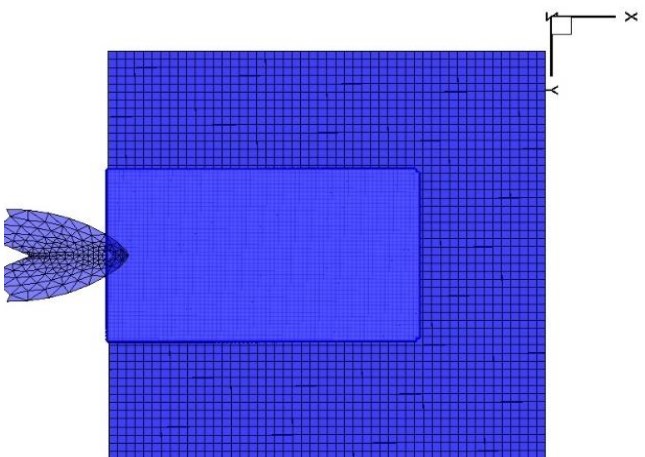

**Figure 10.** Numerical model of ship-ice interaction.

Xuelong icebreaker is selected and its bow is modeled. Main parameters are listed in Table 2. The FEM model of the ship's bow consists of 304 triangular elements. The ship is treated as a rigid body sailing in a straight line at a speed of 3 kn. The ice constitutive model is elastic-brittle, as described in Section 5.1.1. The ice sheet is a rectangle whose edges are fixed, except the one interacting with the icebreaker. The parameters of the ice sheet are presented in Table 3, and the critical bond stretch is calculated using Equation (18). The level ice is modeled using the proposed PD-FEM coupling approach, with PD and FEM subregions. In the PD subregion (width = 40 m, length = 70 m), the grid spacing is $\Delta x = 0.25$ m and the horizon radius is $\delta = 3\Delta x$. In the interface between the PD and FEM subregions, three layers of peridynamics nodes are embedded in each edge. Therefore, there are approximately 187,840 PD nodes. The FEM subregion is discretized into 1800 hexahedral elements of size $2 \times 2 \times 1$ m and 3936 nodes. The size of time step is $\Delta t = 1.0 \times 10^{-7}$ s, which is satisfies with the stability time step condition. The total time steps are 300,000,000 steps.

### 5.1.5. Numerical Result and Discussion

The numerical simulation of the ice-breaking process of an icebreaker navigating through level ice is illustrated in Figure 11. Xuelong model test is performed in the ice mechanics and ice engineering laboratory of Tianjin University to observe the failure model of level ice and the motion of broken ice. After the test is finished, Xuelong model is dragged to slowly retreat by the main trailer, and then a more complete ice breaking area can be shown on the water surface.

**Table 2.** Main parameters of icebreaker.

| Parameter | Variable | Value |
|---|---|---|
| Ship length | $L$ | 166.0 m |
| Ship breadth | $B$ | 22.6 m |
| Ship depth | $D$ | 13.5 m |
| Bow length | $l$ | 29.6 m |
| Bow breadth | $b$ | 22.6 m |
| Draft | $T$ | 8.0 m |
| Stem angle | $\alpha$ | 20° |
| Flooding angle | $\beta$ | 24° |
| Ship-ice friction coefficient | $\mu$ | 0.15 |

**Table 3.** Parameters of level ice.

| Parameter | Variable | Value |
|---|---|---|
| Young's modulus | $E$ | 6.83 GPa |
| Poisson's ratio | $\nu$ | 0.25 |
| Bending strength | $\sigma_f$ | 2.96 MPa |
| Fracture toughness | $K_I$ | 115 kNm$^{-3/2}$ |
| Density | $\rho$ | 894 kg/m$^3$ |
| Area | $A$ | $100 \times 100$ m$^2$ |
| Thickness | $h$ | 1.0 m |

When the icebreaker first contacts the ice, the ice is subjected to a compressive force from the bow tip, mainly along the *x*-axis. Once the force is greater than the ice critical strength, ice particles fall from the level ice, and a notch like the tip of the bow is formed (see Figure 11a). As the ship moves, the notch expands. The three-dimensional curved hull makes the ice bear the force in three directions. The force along the *y*-axis causes the ice to tear, and the notch along the profile forms a radial crack, as shown in Figure 11b.

As the contact area increases, the force along the *z*-axis exceeds the ice critical strength, resulting in ice sheet bending deformation and failure. Accordingly, circular cracks form, as shown in Figure 11c; these are like the circumferential cracks in model test, as shown in Figure 12.

With the ship continuous movement, the contact force increases in all three directions and radial cracks and circular cracks propagate, which is in good agreement with the ice sheet failure model observed in Xuelong model test as shown in Figure 12. The front segments of the level ice form a wedge shape, as shown in Figure 11d. The further movement of the ship causes wedge ice to form on the two sides of the bow, like the real phenomenon observed in Xuelong model test. Subsequently, ice blocks fall off, flip, push away, and pile up, as shown in Figure 11g,h.

Numerical simulation results in the generation and propagation of radial and circular cracks, as well as phenomena such as the shedding of wedge ice, flipping of brash ice, and cleaning of the channel, which are in broad agreement with experimental and real phenomena. In addition, the coupling method of finite element and perdynamics can effectively suppress the boundary effect when the level ice is failure, compared with bond-based peridynamics [28–37].

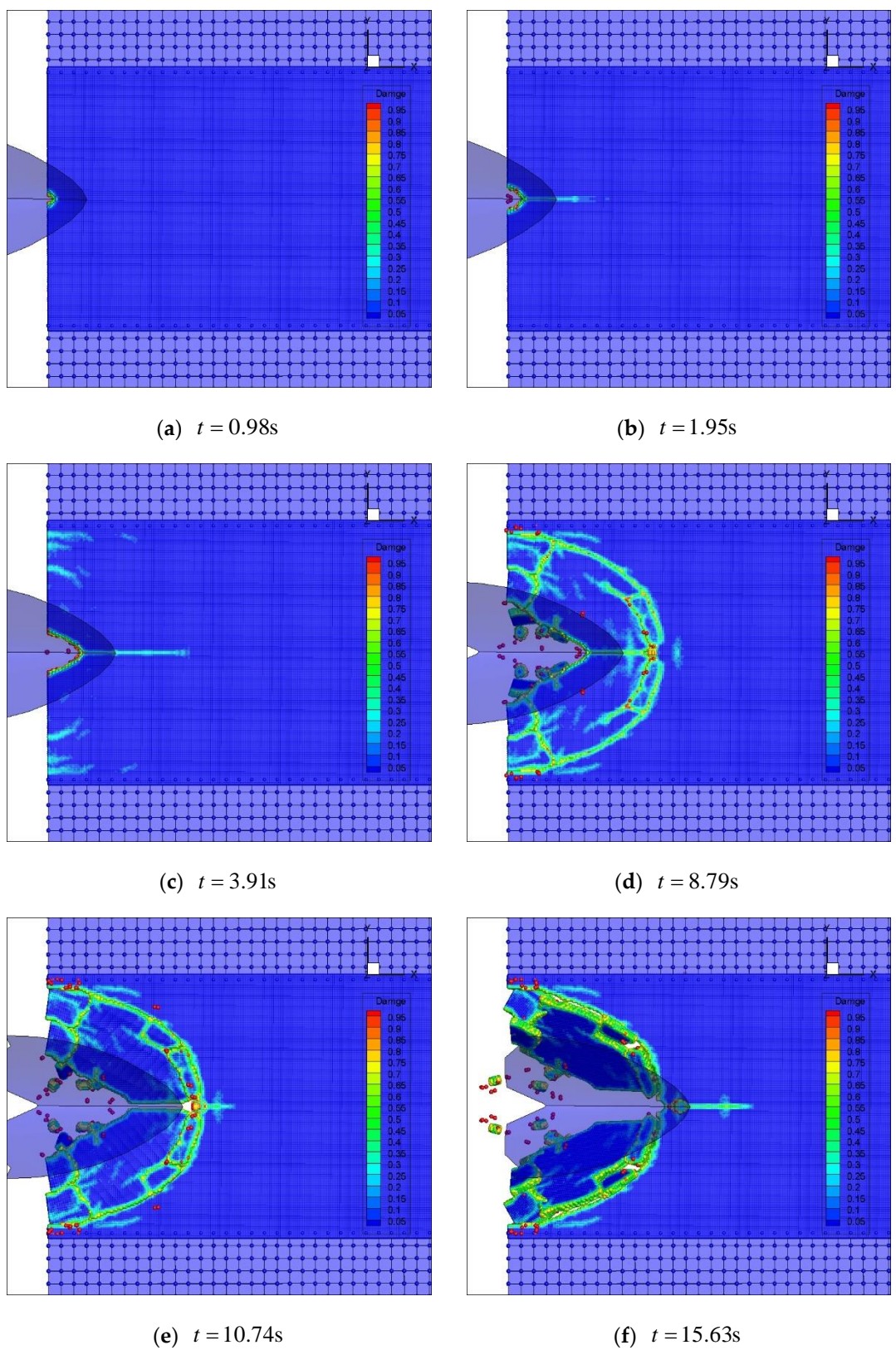

(**a**)  $t = 0.98$s

(**b**)  $t = 1.95$s

(**c**)  $t = 3.91$s

(**d**)  $t = 8.79$s

(**e**)  $t = 10.74$s

(**f**)  $t = 15.63$s

**Figure 11.** *Cont.*

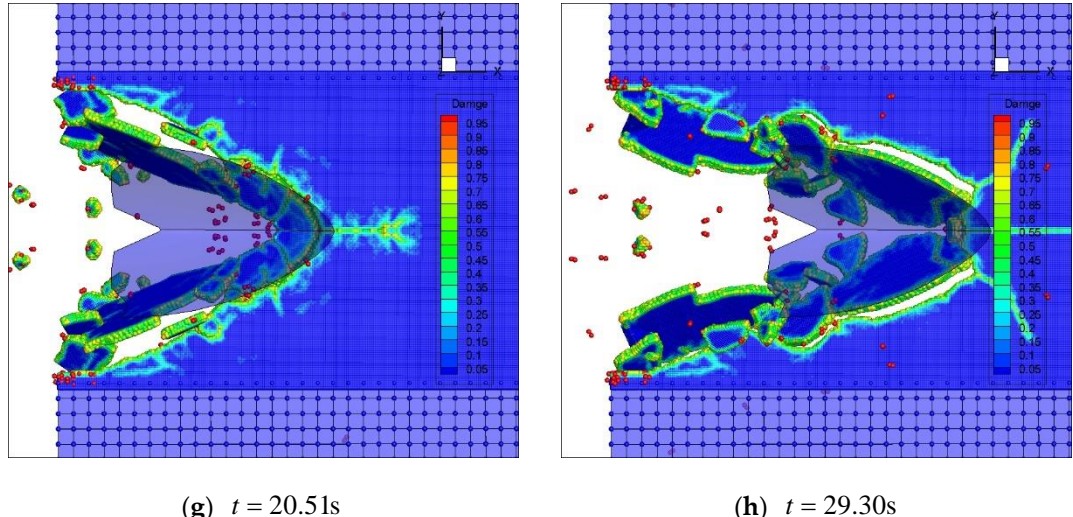

(**g**)  $t = 20.51$s          (**h**)  $t = 29.30$s

**Figure 11.** Ice-breaking process of icebreaker navigating in level ice simulated by PD-FEM coupling model.

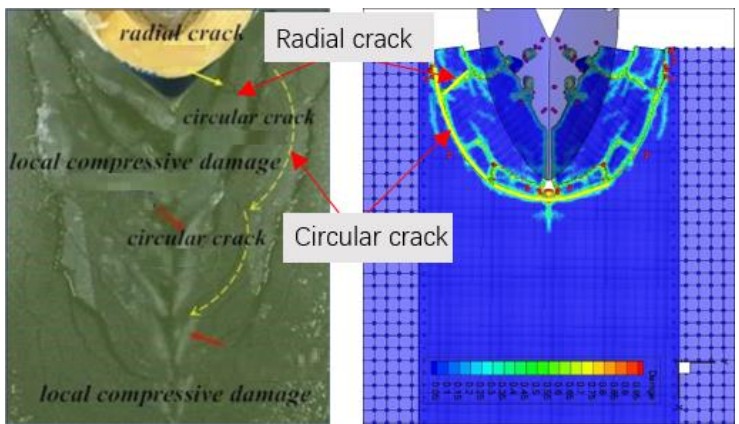

**Figure 12.** Picture of level ice failure in the model test of Xuelong model at Tianjin University.

To show the computational efficiency of PD-FEM coupling approach, the PD solution is considered. Except the level ice is completely modeled by PD solution, loading conditions are the same as the PD-FEM coupling approach. Table 4 shows the comparison results of PD and PD-FEM coupling method in computational time. Both two methods are compiled using Fortran 90, and use CPU_TIME function in Fortran language to calculate the program running time. In the case of the same equipment condition and 300,000 steps of the calculation time steps, the PD method needs calculate 135.47 h, while the PD-FEM coupling approach only needs 52.32 h. Compared with PD model, PD-FEM coupling model is 2.59 times more efficient and reduces the calculation time.

**Table 4.** Comparison of PD and PD-FEM coupling approach in computational efficiency.

| Item | PD-FEM | PD |
|---|---|---|
| Particle number | 187,840 | 654,400 |
| Element number | 1800 | 0 |
| Total time steps | 300,000 | 300,000 |
| Total CPU time | 52.32 h | 135.47 h |

### 5.2. Influence of Ship Speed on Ice Load

To further validate sailing speed influence on the ice load, six simulation sets are selected as speeds with 2, 3, 4, 5, 6 and 7 kn with an ice thickness of 1 m. The ice load results are shown in Figure 13. The average ice force during the period of ship-ice interaction is calculated and compared with Lindqvist's empirical formula as shown in Figure 14.

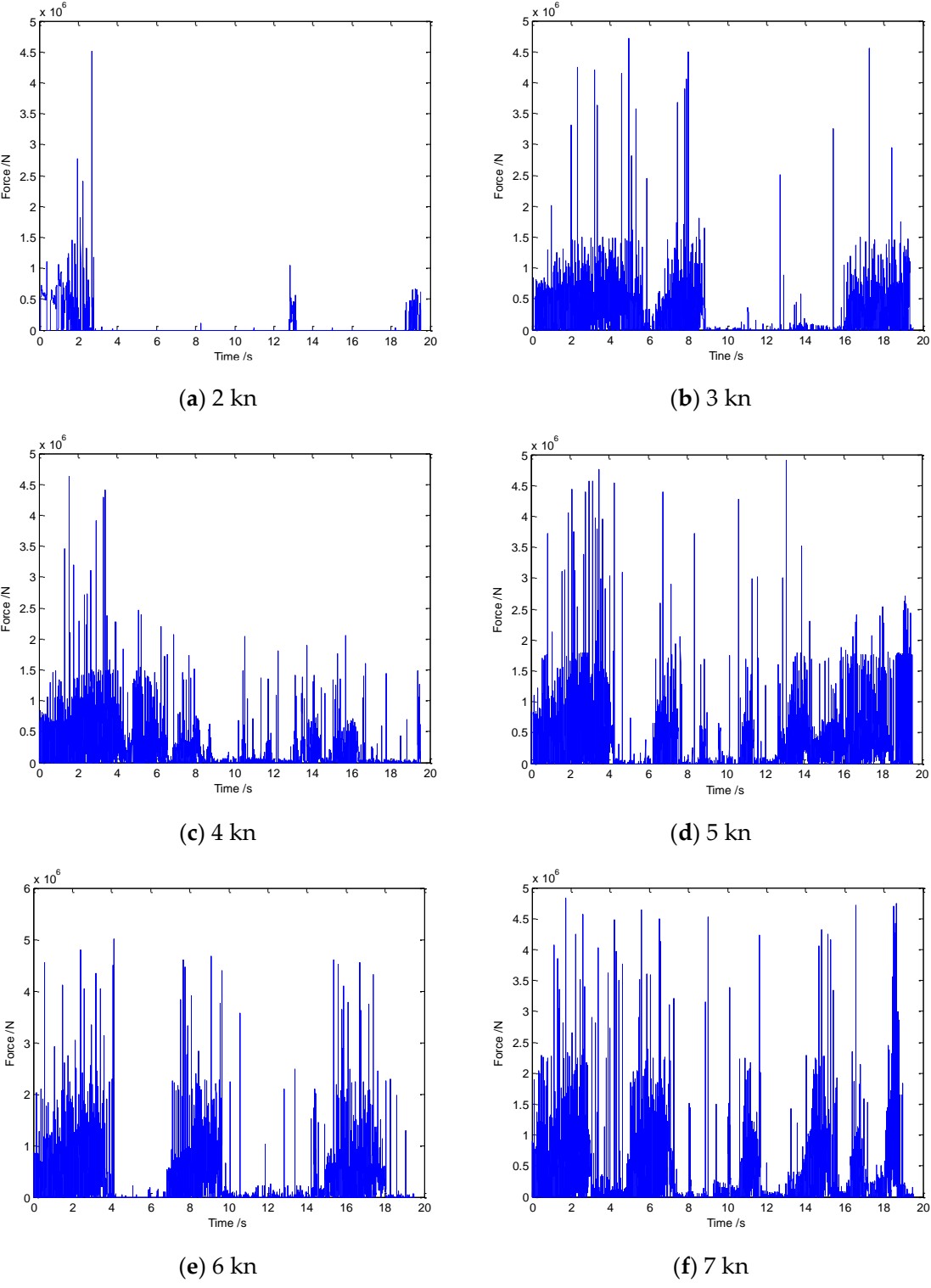

(**a**) 2 kn

(**b**) 3 kn

(**c**) 4 kn

(**d**) 5 kn

(**e**) 6 kn

(**f**) 7 kn

**Figure 13.** Time history of ice load with different ship velocity.

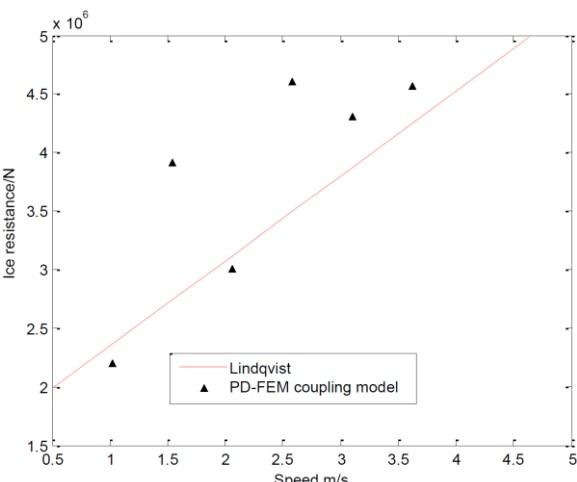

**Figure 14.** Ice resistance obtained by Lindqvist formula and simulated with FEM-PD coupling model for different speeds.

Lindqvist's empirical formula divides the icebreaking resistance into crushing at the stem, breaking by bending and submersion resistance. When ice is broken, the crushed ice can be cleared from both sides of the ship, and the opened channel will be larger than the ship's width. Therefore, the ice crushing at the stem and breaking by bending parts of ice resistance are mainly caused by the bow. Therefore, the ice load obtained from numerical simulation is compared with the crushing at the stem and breaking by bending resistance calculated by Lindqvist's empirical formula.

Figure 13 shows that the ice load curves (blue solid lines) are periodic. When the icebreaker interacts with the level ice, the ice load increases, but when wedge ice falls from the level ice and the icebreaker is not in contact with the ice, the ice load decreases. This is a reason for impulsive ice loads. Figure 13 also indicates that, as the ship speed increases, the period of the ice load becomes shorter and the ship velocity influences the peak of ice load. The ice load shows a rise trend as the velocity increases. From Figure 14, although there are some differences between the mean ice load results and those obtained from Lindqvist's empirical formula, they are generally in a good agreement (see Table 5 and Figure 14).

**Table 5.** Ice resistance calculated by PD-FEM coupling model and Lindqvist empirical formula.

| Method | 2 kn | 3 kn | 4 kn | 5 kn | 6 kn | 7 kn |
|---|---|---|---|---|---|---|
| Lindqvist | $2.370 \times 10^6$ N | $2.743 \times 10^6$ N | $3.120 \times 10^6$ N | $3.490 \times 10^6$ N | $3.863 \times 10^6$ N | $4.237 \times 10^6$ N |
| PD-FEM | $2.202 \times 10^6$ N | $3.921 \times 10^6$ N | $3.008 \times 10^6$ N | $4.612 \times 10^6$ N | $4.307 \times 10^6$ N | $4.560 \times 10^6$ N |
| Errors | 7.1% | 42.9% | 3.6% | 32.1% | 11.5% | 7.6% |

*5.3. Influence of Ice Thickness on Ice Load*

Numerical simulations of an icebreaker moving at a fixed speed are performed with different ice thicknesses. The ice thickness is selected as 0.5 m, 0.75 m, and 1 m, and the ship's speed is set to 3 kn.

The results of the ice load for different ice thicknesses are shown in Figure 15. It can be found that ice load presents an increasing trend in general. The mean ice load is 0.978 MN, 1.569 MN and 3.921 MN at ice thicknesses of 0.5 m, 0.75 m and 1.0 m, respectively. From Figure 16, we can find that the numerical results are in well agreement with those from Lindqvist's empirical formula.

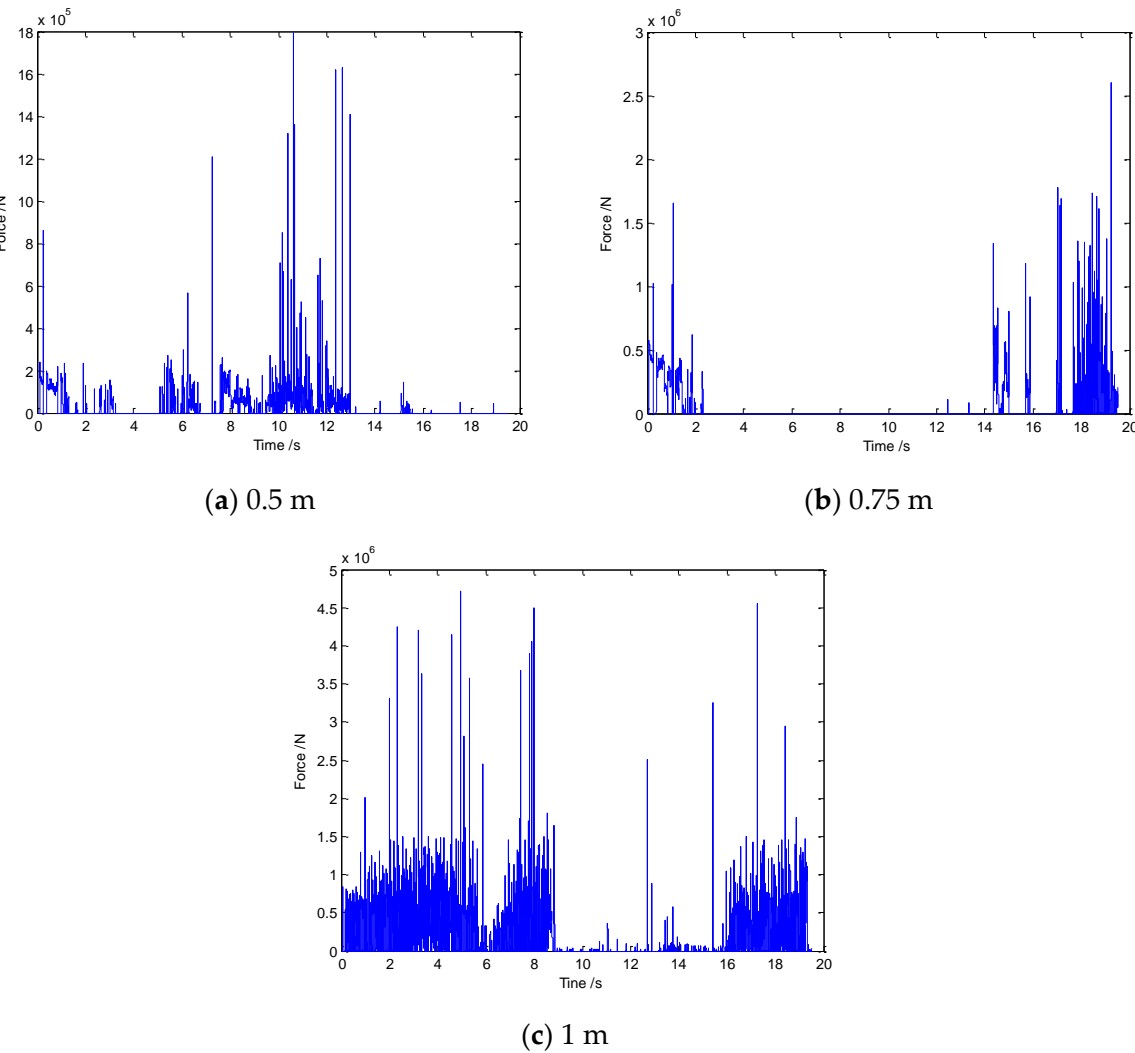

(**a**) 0.5 m

(**b**) 0.75 m

(**c**) 1 m

**Figure 15.** Time history of ice load with different ice thicknesses.

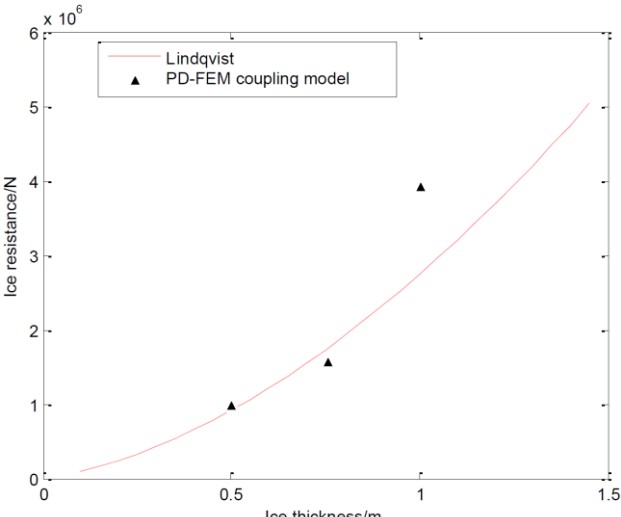

**Figure 16.** Ice resistance obtain by Lindqvist formula and FEM-PD coupling model for different ice thicknesses.

## 6. Conclusions

The coupling model of peridynamics with the finite element method is employed to simulate ship–ice interaction. The characteristics of the ice-breaking scenarios and the ice load are captured successfully. From the simulation results, the following conclusions can be drawn.

(1) The PD-FEM coupling model can successfully simulate the generation and propagation of radial and circular cracks in level ice, as well as the phenomena of wedge ice shedding, broken ice flipping, and ice cleaning of the channel during the ice-breaking process.

(2) Compared with bond-based peridynamics, the PD-FEM coupling model has better computational efficiency, and can effectively suppress the boundary effect when the level ice is failure.

(3) The ice load obtained from the PD-FEM coupling model is in good agreement with that obtained from Lindqvist's empirical formula.

**Author Contributions:** Conceptualization, Y.X., R.L. and X.L.; methodology, Y.X. and X.L.; program, R.L. and X.L.; validation, Y.X., R.L. and X.L.; investigation, R.L. and X.L.; writing—original draft preparation, Y.X. and R.L.; writing—review and editing, Y.X., R.L. and X.L.; supervision, Y.X. All authors have read and agreed to the published version of the manuscript.

**Funding:** This research was funded by the National Natural Science Foundation of China, Grant No. 51979056.

**Institutional Review Board Statement:** Not applicable.

**Informed Consent Statement:** Not applicable.

**Data Availability Statement:** The data presented in this study are available on request from the corresponding author. The data are not publicly available due to privacy.

**Conflicts of Interest:** The authors declare no conflict of interest.

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
