# Peer review of "Coupling of Finite Element Method and Peridynamics to Simulate Ship-Ice Interaction"

_jmse, doi:10.3390/jmse11030481_

Round 1

Reviewer 1 Report

This paper provides us the important information to develop the coupling technique between FEM and recently fashioned peridynamics(PD).

The idea to start the topic is quite excellent. But, unfortunately, the descriptions are so insufficient to understand.

Therefore, I achieve the conclusions not to recommend this paper to the publication for JMSE in the  present form.

The following comments are useful for the revision.

1. The overlapping two regions should be careful more. Many trials should be made by changing delta as well as delta x.

2. Related to that, the author refer the appropriate delta = 3 delta x from the book published 8 years ago. If this is correct, this kind of approaches has already been done before 8 years. This means the originality of this paper is quite low and there is nothing new.

3. I have a doubt on Eq. (6). Basically, shape function is defined for a discretization of displacement. Without any checks by changing the interpolation functions and their order, I do not agree it is effective.

4. The authors have no ideas to coincide the position of the particle for PD to the nodes on FEM. The reasons should be explained.

5. If the problem of the ice breaking by a ship discussed in the section 5 will be simulated only by explicit FEM (e-FEM) with the element erosion technique, I do really would like to know what kind of the results will be obtained.

6. PD is quite advantageous for reproducing the crack branching phenomena. The straight extension of single crack in 2D space under tension is not good example. Related to the above, the comparison of the result with only PD as well as the element erosion technique by e-FEM is quite interesting. The reliability and efficiency of the proposed technique will be enhanced.

7. This kind of overlapping technique between FEM and the particle has already been done in the past score. Especially, a coupling technique between the atomistic simulation has already been challenged. Not only the molecular dynamics but also the particle simulation such as SPH, MPS or the other meshless techniques have already made. I think it is better to refer them and improve the simulation by inclusions of their ideas.

8. Totally, I do not find the advantage to couple between two method. The advantages should be emphasized in the text.

Author Response

Reply to the questions of Reviewer No. 1

Reviewer #1: This paper provides us the important information to develop the coupling technique between FEM and recently fashioned peridynamics(PD). The idea to start the topic is quite excellent. But, unfortunately, the descriptions are so insufficient to understand. Therefore, I achieve the conclusions not to recommend this paper to the publication for JMSE in the present form.

The following comments are useful for the revision.

Q1. The overlapping two regions should be careful more. Many trials should be made by changing delta as well as delta x.

R1: We thank the reviewer for this comment.

δ - convergence and m - convergence are essential prior to numerical simulations. For the current bond-based peridynamics, it is well known that m = 3 can get acceptable result without significantly computational time (Xue Y, Liu R and Liu Y, et al.). To check the effect of horizon size, more cases were done in the updated manuscript. The results are shown in Fig. 5. We can find that the horizon size has little influence on the displacement if the horizon is less than the minimum calculation direction.

Xue Y, Liu R and Liu Y, et al. Numerical simulations of the ice load of a ship navigating in level ice using peridynamics[J]. Computer Modeling in Engineering, 2019: 523-550.

Q2. Related to that, the author refer the appropriate delta = 3 delta x from the book published 8 years ago. If this is correct, this kind of approaches has already been done before 8 years. This means the originality of this paper is quite low and there is nothing new.

R2: We would like to thank the reviewer for this comment.

Actually, the coupling strategy used in the current model is borrowed from the work done by Liu W. and Hong J.W. The work in this manuscript is focus on the engineering applications of this coupling technique. We also find that the influence from boundaries can be avoid in the current model, compared with the work (Xue Y, Liu R and Liu Y, et al.)..

Liu, W. and Hong, J.W., 2012. A coupling approach of discretized peridynamics with finite element method. Computer methods in applied mechanics and engineering, 245, pp.163-175.

Xue Y, Liu R and Liu Y, et al. Numerical simulations of the ice load of a ship navigating in level ice using peridynamics[J]. Computer Modeling in Engineering, 2019: 523-550.

Q3. I have a doubt on Eq. (6). Basically, shape function is defined for a discretization of displacement. Without any checks by changing the interpolation functions and their order, I do not agree it is effective.

R3: We thank the reviewer for this comment.

We apologize for this confusion. The Eq.(6) was written in a wrong way, and it should be

We can find more details in the work by Liu W. and Hong J.W.

Liu, W. and Hong, J.W., 2012. A coupling approach of discretized peridynamics with finite element method. Computer methods in applied mechanics and engineering, 245, pp.163-175.

Q4. The authors have no ideas to coincide the position of the particle for PD to the nodes on FEM. The reasons should be explained.

R4: We thank you for this comment.

The PD-FEM coupling approach proposed by Liu et al.[35] is adopted in the current work. The position of the particle for PD coinciding with the nodes on FEM It is consistent to the original work. We think that one of the advantages of doing this is to simplify the model and calculation of particles (nodes) information in the coupling area.

Q5. If the problem of the ice breaking by a ship discussed in the section 5 will be simulated only by explicit FEM (e-FEM) with the element erosion technique, I do really would like to know what kind of the results will be obtained.

R5: We thank you for this comment.

Numerical models based on the FEM to predict ship-ice interactions and ice loads can be referred to [Han, Feng and Yue (2007); Kolari, Kuutti and Kurkela (2009); Nylandsted, Jäättelä, Hoffmann et al. (2003); Premachandran and Horii (1994); Ranta, Polojärvi and Tuhkuri (2018); Xue (2016)]. Compared with these FEM results, we can find that characteristics of the icebreaking process can be better obtained using the peridynamic model such as the dynamic generation of cracks in the ice sheet, propagation and accumulation of ice fragments, as well as collision, rotation, and sliding of the ice fragments along the ship hull.

Han, L.; Feng, L. I.; Yue, Q. (2007): Simulation of the whole failure process by FEM during ice-conical structures interaction. China Offshore Platform.

Kolari, K.; Kuutti, J.; Kurkela, J. (2009). FE-simulation of continuous ice failure based on model update technique. Proceedings of the International Conference on Port and Ocean Engineering Under Arctic Conditions.

Nylandsted, J.; Jäättelä, M.; Hoffmann, E. K.; Pedersen, S. F. (2003): Finite element modelling and simulation of indentation testing: a bibliography (1990-2002). Engineering Computations, vol. 21, no. 1, pp. 23-52, 30.

Premachandran, R.; Horii, H. (1994): A micromechanics-based constitutive model of polycrystalline ice and FEM analysis for prediction of ice forces. Cold Regions Science and Technology, vol. 23, no. 1, pp. 19-39.

Ranta, J.; Polojärvi, A.; Tuhkuri, J. (2018): Ice loads on inclined marine structures-Virtual experiments on ice failure process evolution. Marine Structures, vol. 57, pp. 72-86.

Xue, H. (2016). Investigation of ice-PVC separation under flexural loading using FEM analysis. International Journal of Multiphysics, vol. 10, no. 3, pp. 247-264.

Q6. PD is quite advantageous for reproducing the crack branching phenomena. The straight extension of single crack in 2D space under tension is not good example. Related to the above, the comparison of the result with only PD as well as the element erosion technique by e-FEM is quite interesting. The reliability and efficiency of the proposed technique will be enhanced.

R6: We thank the reviewer for this suggestion.

The 2D crack case is mainly used to illustrate the feasibility of the current coupling method, and also to verify the correctness of the self-programming model. For this work, the main purpose is to enrich the existing ship-ice interaction simulation method. To compare with e-FEM, it adds a lot of work and this article may not be able to solve it. We will compare the simulation results of these two methods in the future.

Q7. This kind of overlapping technique between FEM and the particle has already been done in the past score. Especially, a coupling technique between the atomistic simulation has already been challenged. Not only the molecular dynamics but also the particle simulation such as SPH, MPS or the other meshless techniques have already made. I think it is better to refer them and improve the simulation by inclusions of their ideas.

R7: We thank the reviewer for this comment.

The current coupling method was proposed by Liu et al. in 2012, and it is borrowed in this work. We original purpose is to enrich the existing ship-ice interaction simulation method.

Liu, W. and Hong, J.W., 2012. A coupling approach of discretized peridynamics with finite element method. Computer methods in applied mechanics and engineering, 245, pp.163-175.

Q8. Totally, I do not find the advantage to couple between two method. The advantages should be emphasized in the text.

R8: We thank the reviewer for this comment.

There are two advantages. One is that the area without permitting damage is calculated by FEM, which can solve the calculation time. The other is that by comparing with the bond-based PD method, we can find that the boundary effect on level ice damage has been effectively improved.

Reviewer 2 Report

Please improve the following parts of the manuscript prior to its publication:

- There are many attempts using DEM, SPH, and other methods. Introduction should also list these recent attempts.

- One paragraph in the Introduction section should describe the advantages of the proposed methodology, compared to other modern coupling methods.

- Mesh dependency should be introduced. How can the validation be certain for only one mesh size? Please introduce another coarser simulation results (coarser FE mesh, and coarser PD spacing).

- Fig 12. is not directly comparable to numerical results. Please include side-by-side comparison and point to corresponding regions.

- For a continuous numerical methodology, Figure 13 is depicting a very discrete and spiky force output. This reviewer has not found the explanation for it in the manuscript. Please explain why it happens and why is it okay.

- Connected to the above point, why the authors have not analyzed the impulse (the force integral as well), as it is common practice for such kinds of FSI?

Author Response

                                               Reply to Reviewer No. 2

Reviewer #2: Please improve the following parts of the manuscript prior to its publication:

Q1. There are many attempts using DEM, SPH, and other methods. Introduction should also list these recent attempts.

R1: We would like to thank the reviewer for this suggestion.

We added essential references about DEM, SPH, CZM in the introduction.

Q2. One paragraph in the Introduction section should describe the advantages of the proposed methodology, compared to other modern coupling methods.

R2: We thank the reviewer for this comment.

We added the advantages of the coupling model with other coupling methods.

Q3. Mesh dependency should be introduced. How can the validation be certain for only one mesh size? Please introduce another coarser simulation results (coarser FE mesh, and coarser PD spacing).

R3: We thank the reviewer for this comment.

Convergence investigation referred to the work done in Xue et al, and the mesh size is selected based on it. The convergences are essential before any simulation applications. We added more cases on the influence of horizon size to validate the values used in the current work is acceptable. We also try to run more cases about ship-ice interaction, but one case takes more than 10 days. At the time of submission, the simulation is still running.

Xue Y, Liu R and Liu Y, et al. Numerical simulations of the ice load of a ship navigating in level ice using peridynamics[J]. Computer Modeling in Engineering, 2019: 523-550.

Q4. Fig 12. is not directly comparable to numerical results. Please include side-by-side comparison and point to corresponding regions.

R4: We would like to thank the reviewer for this comment.

We added side-by-side comparison in the updated manuscript.

Q5. For a continuous numerical methodology, Figure 13 is depicting a very discrete and spiky force output. This reviewer has not found the explanation for it in the manuscript. Please explain why it happens and why is it okay.

R5: We thank the reviewer for this comment.

We are sorry for this confusion. Due to the simplification of the fluid with an elastic foundation, the level ice deforms and failures without sufficient support. Ice block will be larger than the normal observation during ship-ice interaction. Because the ice force only occurs in the process of ship-ice interaction, the large ice blocks lead to less ship ice collision times, so the output results show the phenomenon of the article.

To make the result compared with Lindqvist’s empirical formula, the ice load obtained from numerical simulation is only compared with the crushing at the stem and breaking by bending resistance calculated by Lindqvist’s empirical formula.

Q6. Connected to the above point, why the authors have not analyzed the impulse (the force integral as well), as it is common practice for such kinds of FSI?
R6: We thank the reviewer for this suggestion.

The purpose of this work is to present a coupling approach incorporating FEM and PD to simulate ship-ice interaction, and the hull is treated as a rigid body without any deformation. We are also working on the ice force characteristics and simulating the ship with elastoplastic deformation.

Reviewer 3 Report

Comments in attachment

Author Response

                                               Reply to Reviewer No. 3

Reviewer #3: Comments to authors: An interesting work and a well written paper.

Q1. About the comparison/benchmark as presented in table 1 and figure 5.

Comparing pd -fem model with the theoretical solution does not only show the effects of coupling in the pd-fem model. It is suggested that the authors compare the results for the PD-FEM model also with the FEMFEM model (in Figure 4) and include these results in Table 1 and Figure 5. The coupling effects with respect to force and displacements should then be directly revealed.

R1: We would like to thank the reviewer for this comment.

We are sorry for this confusion. The theoretical solution should be replaced by FEM, and we mixture these concepts.

Q2. Suggestions for edits:

1) Figure caption for figure 7: Missing text for the left and the right column.

2a) "The time step is …..s, which is satisfied with the stable time step condition." should be

rewritten e.g . The time step is ... which satisfies the stable time step condition

2b) On line 338: as with 2a

3) Repeated text on line 138-139

4) Table 4: PD and PD-FEM has been interchanged in the heading

R2: We thank the reviewer for this comment.

We are sorry for these confusions and mistakes. We have checked the manuscript and modified similar problems. All the changing things are marked in the updated manuscript.

Round 2

Reviewer 1 Report

Thank the authors very much for spending time to improve the manuscript.

I hope the authors should reflect the comments into the text. I do not understand where the improvement by my comment was made. In the answer sheets, the specification of the improvement should be indicated.

Reviewer 2 Report

The authors have revised all neccesary details, and the manuscript is ready for publication.